# How exclusion criteria can hinder eligibility for lung cancer studies among different racial and ethnic groups

Jennifer Y. Kim[1,*], Abigail Dirks[1], Ruby Madison Ford[1], Lori Pai[2], Calvin Ludwig[3], Umit Tapan[4]

**1** Tufts Center for the Study of Drug Development, Tufts School of Medicine, Boston, Massachusetts, United States of America, **2** Hematology Oncology, Hawaii Pacific Health Straub Medical Center, Honolulu, Hawaii, United States of America, **3** Internal Medicine, Tufts Medical Center, Tufts School of Medicine, Boston, Massachusetts, United States of America, **4** Hematology & Medical Oncology, Boston Medical Center, Boston University School of Medicine, Boston, Massachusetts, United States of America

* Jennifer_y.kim@tufts.edu

## Abstract

The extent to which protocol eligibility criteria contribute to the underrepresentation of racial and ethnic minority populations — including Black, Asian, and Latino Americans — in lung cancer clinical trials remains poorly characterized. This study quantifies the likelihood of clinical trial exclusion attributable to comorbid conditions across racial and ethnic groups among patients with lung cancer. Data were drawn from 1,134 lung cancer clinical trials registered on ClinicalTrials.gov with start dates between January 2014 and December 2024, and patient comorbidity data were obtained from electronic medical records (EMR) at a large urban academic medical center in the Northeast United States. Data analysis was conducted between February and May 2025. Eligibility for trial enrollment was assessed by mapping patient comorbidity profiles against study exclusion criteria; binary logistic regression was used to estimate the likelihood of exclusion by race and ethnicity, with sex and median household income included as covariates. The analytic sample comprised 4,096 patients with lung cancer (73.6% White, 12.8% Asian or Pacific Islander, 3.3% Black or African American, and 1.8% Hispanic/Latino). Compared to White American patients, Asian American and Pacific Islander (AAPI) patients and Black or African American patients were 1.8 times (OR: 1.8, 95% CI: 1.03–3.03) and 1.6 times (OR: 1.6, 95% CI: 1.01–2.48) more likely to be excluded from clinical trials based on their comorbidities, respectively. These findings indicate that standard protocol exclusion criteria may disproportionately screen out racial and ethnic minority patients, particularly Black/African American and AAPI individuals, and may represent a structural contributor to their underrepresentation in lung cancer research. Revising eligibility criteria to better reflect real-world comorbidity burdens could improve the inclusivity and generalizability of lung cancer clinical trials.

**Data availability statement:** Because the dataset consists of individual-level electronic medical record (EMR) data containing potentially identifiable health information, it cannot be made publicly available. However, access to de-identified data may be granted to qualified researchers upon reasonable request, subject to institutional data use agreements and applicable regulatory and ethical approvals including Tufts Medical Center IRB (IRBOffice@tuftsmedicine.org) and the Office of the Vice Provost for Research (OVPR@tufts.edu).

**Funding:** The work was supported by funding from Novartis Investigator Initiated Research (grant ID: PTO300) (JYK, AD, and RMF). This content is solely the responsibility of the authors and does not necessarily represent the official views of the funder. The funders had no role in study design, data collection and analysis, decision to publish, or preparation of the manuscript. Website: https://www.novartis.com/.

**Competing interests:** The authors have declared that no competing interests exist.

## Author summary

The development of treatments for patients with lung cancer relies on the enrollment of patients in clinical trials; however, racial and ethnic minority patients have remained underrepresented in these studies, raising questions about the efficacy and safety of treatments for underenrolled patient populations. While several factors contribute to the under enrollment, one factor that may hinder accessibility are stringent protocol exclusion criteria. For example, certain medical conditions that have a higher incidence among racial and ethnic minority patients may be listed as exclusion criteria, preventing patients from these communities from qualifying for lung cancer studies. The extent to which medical conditions listed under exclusion criteria can impede eligibility for patients into lung cancer clinical trials remains largely unknown. To fill this gap, we used patient electronic medical records (EMR) and mapped them to exclusion criteria taken from lung cancer clinical trial studies registered on ClinicalTrials.gov. We then quantified the odds that a patient would qualify for a lung cancer study based on their medical condition. Results showed that Asian American and Pacific Islander (AAPI) patients and Black or African American patients were significantly more likely to be excluded from clinical trials than White American patients based on their comorbidities. We discuss implications of our study results, adding to the existing research on reducing clinical trial protocol complexity to make lung cancer studies more accessible to patients.

## Introduction

Participating in clinical trials improves patients' access to new treatments and enhances health outcomes, particularly for diseases with high mortality rates [1]. However, in the United States (U.S.), enrollment of racial and ethnic minority populations—including Black, Asian, and Latino Americans—remains low in oncology research, including lung cancer studies [2]. The lack in representation is problematic because lung cancer continues to disproportionately affect racial minority populations: for example, Black American men are 15% more likely to develop the disease compared to White men [3]; Hispanic Americans are 30% more likely to not receive any treatment compared to White patients [4]; and Asian Americans are 17% less likely to be diagnosed early than White patients. Despite being disproportionately affected by lung cancer, Black, Latino, and Asian Americans continue to remain underrepresented in lung cancer clinical trials, resulting in studies that lack drug safety profile data for different racial and ethnic minority groups and raising questions about the safety and efficacy of these treatments [5].

One contributing factor to the underrepresentation of racial minority populations in lung cancer clinical trials is the increasing complexity and stringency of trial protocols, which can make referral and enrollment into these studies challenging. Indeed, this pattern has been well documented in the lung cancer research [6,7] Exclusion criteria

often list medical conditions—such as chronic disorders including diabetes and heart disease —that are more prevalent and concentrated among racial and ethnic minority groups than the general population, limiting their eligibility [8,9]. Existing research using patient reported data suggests that the presence of one or more comorbidities is associated with lower chance of trial discussion and trial participation among patients with cancer [10]. Further, some criteria rely more heavily on subjective language—such as "physician determines that a patient will not adhere to the study protocol"— than on more objective clinical observations, increasing the potential for bias to influence an investigator's decision when recruiting patients who do not fit the ideal patient prototype, which is typically college-educated, middle to high income, White Americans [11,12].

Taken together, the current state of lung cancer eligibility criteria can pose serious roadblocks that can hinder trial participation among different racial and ethnic groups, especially among patients who have chronic comorbidities, highlighting a need for revising and modernizing protocols. This study aims to quantify the direct impact that current protocol eligibility requirements based on comorbidities can have on patient eligibility and determine whether study criteria adversely exclude different racial and ethnic groups based on their comorbidity status. We do so by combining lung cancer data from ClinicalTrials.gov and patient electronic medical records (EMR) from a single, large-sized urban academic medical center in the Northeast USA. Specifically, we generated a large, triangulated dataset, using patient data to examine whether lung cancer studies exclude different patient groups based on their comorbidities and calculate the odds that a patient would be disqualified from a lung cancer study based on their preexisting conditions.

## Methods

The exclusion criteria dataset was built using study data obtained from ClinicalTrials.gov's registered trial database. An Application Programming Interface (API) query extracted all U.S.-based lung cancer trials with trial start dates from January 2014 to December 2024, resulting in 1,134 studies. See S1 eAppendix in S1 File for list of search terms that were used to capture lung cancer studies. To account for the fact that not all exclusion criteria are based on a patient's medical condition/history, we classified the different criteria into 5 categories: 1) Medical Condition or Medical History, 2) Medical Treatment, 3) Clinical Exclusion, 4) Gatekeeper Bias, and 5) Inability to Consent. Table 1 lists the definition of each of the 5 categories and a sample of the keywords used to identify each.

A list of the common terms in the criteria was extracted. Using these terms, string matching was used to label each exclusion criterion. An iterative process was then conducted with manual checking done by the research team. To ensure calibration, the research team manually checked a random sample of 1,000 out of 13,759 categorized criteria, and adjustments to the regular expressions were made accordingly, including additional keywords and patterns that were initially missing. For example, if an indicator occurred after the word "except for," then that exclusion criterion was excluded from being classified into that category. An additional manual review was conducted for the criteria that were categorized into more than one category to create orthogonal categories. However, about 16% of the criteria were compounded and were classified into multiple categories. Criteria classified into multiple categories would be included in the patient exclusion analysis if at least one of those categories was medical history/condition. The core team, comprised of the primary investigator, research analyst, and data scientist, regularly consulted with a team of thoracic oncologists to ensure that the classifications were accurate. See S1 eAppendix in S1 File for additional details.

This categorization process was used for the medical history/medical condition category; we aimed to match each of the condition in the exclusion criteria to the diagnosed conditions from the patient EMR data. Patient EMR data was derived from a large urban academic medical center located in the Northeast. A total of 4,096 patients with lung cancer were identified; the sample was 73.6% White, 12.8% Asian, 3.3% Black, and 1.8% Hispanic/Latino. To form a list of criteria labeled as medical condition/history, we identified the most common terms from the clinicaltrial.gov exclusion criteria dataset, defined as single word or two-word phrases that occurred in at least 0.02% of all exclusion criteria in the dataset (around 20 occurrences out of 8398 criteria). The research team reviewed and extracted the medical conditions

**Table 1. General Exclusion Criteria Categories.**

| Category | Definition | Sample Keywords |
|---|---|---|
| Clinical Exclusion | Criteria that exclude patients based on a specific clinical measurement (i.e., laboratory or vital measurements). | Platelet, hemoglobin, weight |
| Medical History/ Condition | Criteria that exclude patients based on a medical diagnosis that they have or have had in the past. | Tumor, disease, disorder |
| Medical Treatment | Criteria that exclude patients based on any medical treatment that they may be undergoing, have undergone, or will undergo over the course of the trial, besides the study drug. | Steroid therapy, previous treatment, radiation |
| Inability to Consent | Criteria that exclude patients who through clinical observations, are deemed to have conditions, such as a cognitive impairment that make them unable to consent based on observations by the provider or investigator. Criteria are very specific about what this entails (less open to subjectivity) | In the opinion of the investigator, unable to consent, legal capacity, prisoner, clinically significant, psychiatric disorder, substance abuse, impaired decision-making, substance abuse |
| Gatekeeper Bias | Criteria that exclude patients based on perceptions of the provider or investigator using subjective language, not based on clinical assessments (more open to subjectivity) | In the opinion of the investigator, unable to comply |

associated with those terms. The team worked with Clinical and Translational Science Institute (CTSI) at Tufts Medical Center to extract EMR dataset of lung cancer patients and their diagnoses based on the list of conditions that were found in the exclusion criteria. The diagnoses also included any subtypes from the Athena Observational Health Data Sciences and Informatics (OHDSI) Vocabularies Repository. To ensure the diagnoses in the EMR data matched the conditions in the clinicaltrials.gov data, the subtypes and variations for specific disease conditions from the Athena OHDSI Vocabularies repository were also used to enhance the string matching of the exclusion criteria data (i.e., essential hypertension or idiopathic hypertension were classified as hypertension) [13]. Systematized Nomenclature of Medicine (SNOMED) codes were used to identify conditions in both datasets.

## Simulation & analysis

Once the medical conditions in the EMR and ClinicalTrials.gov datasets were paired using disease condition as the unique identifier, the two datasets were merged into a long format such that each lung cancer patient's medical condition was compared to each criterion for every trial, resulting in 3,862,528 patient and trial combinations. Patient race, ethnicity, and sex were extracted directly from the EMR dataset. To account for socioeconomic status, patients' home Zip codes were matched with median household income data sourced from the U.S. Census American Community Survey [14].

## Sensitivity analysis

A sensitivity analysis was conducted to assess our study's ability to detect meaningful racial disparities in exclusion rates. Using our observed baseline exclusion rate in White patients, we calculated the minimum detectable effect (MDE) for each racial group with 80% power at $\alpha = 0.05$. Clinically meaningful disparities were defined as: small (5 percentage points, OR≈1.29), medium (10 percentage points, OR≈1.67), and large (15 percentage points, OR≈2.19) increases in exclusion rates (Prior to data collection, a priori power analysis was conducted to determine the minimum sample size required for the logistic regression analysis. Based on the number of predictors (race/ethnicity, sex, and income), a small effect size and power of.95, the analysis indicated a minimum required sample of $n = 395$. To account for potential or incomplete data, an additional 25 participants were added, yielding a minimum size of $n = 420$. We acknowledge that this power calculation assumes equal distribution across racial groups; however, unequal representation across groups may result in reduced power to detect differences for smaller racial subgroups. Indeed, our actual patient population was

4,096, well above the minimum size of n = 420. However, the racial and ethnic groups were unbalanced. To address the underlying concern about sample size adequacy, we performed a retrospective sensitivity analysis to determine the minimum detectable effect (MDE) for each racial group.).

### Main analysis

We used binary logistic regression to examine the association between patients' race and ethnicity and their exclusion from the trial. The outcome variable was coded as 1 if the patient had at least one diagnosis matching the trial's exclusionary criteria, and 0 if the patient had no matching exclusionary diagnosis. Covariates included sex and median household income of the zip code where the patient resides. Logistic regression assumptions were assessed. Multicollinearity was evaluated using the variance inflation factor, which was less than 2 for each predictor. The independence assumption was not met, since each patient and each trial were repeated in the dataset. To adjust for this, standard errors were adjusted to handle clustering. All analyses were conducted using R [15].

### Ethics Statement

The study was reviewed and approved by Tufts University's Institutional Review Board (IRB) under STUDY00004811.

### Results

First, we report the distribution of the different categories of exclusion criteria. The most common exclusion was medical condition/medical history, which is the focus of our current study. More than half (61.6%) of trials also excluded patients based on clinical exclusion criteria, which are criteria that require specific laboratory or vital measurements to be met in order to qualify for a study. Gatekeeper bias occurred in almost 30% of trials, while inability to consent occurred in about 6% of trials. See Table 2.

Next, to assess patient medical comorbidity's association with a patient's eligibility for a clinical trial, a logistic regression model was built where the binary outcome determining a patient's qualification for a lung cancer study (Yes/No) was regressed on the predictors – race and ethnicity, adjusting for sex and median household income. The model revealed that compared to White patients, certain racial minority subgroups were more likely to be excluded from trials based on their comorbidity diagnoses. See Table 3 for the regression output where we report the logistic regression coefficients and the odds ratio, for. Results showed that compared to White patients, Asian American and Pacific Islander (AAPI) patients were 1.76 times more likely to be excluded, and Black or African American patients were 1.59 times more likely to be excluded. Results based on ethnicity (Hispanic or Latino), sex, and median household income were not significant.

**Sensitivity Analysis.** Baseline exclusion rate in White patients (36% from the logistic regression model, Table 3) were used to determine the minimum detectable effect (MDE) for each racial group. Results from the sensitivity analysis showed that the minimum detectable effects were: AAPI (n = 512): 6.4 pp; Black (n = 132): 11.8 pp; and Latino (n = 72): 15.8

**Table 2. Types of Exclusion Criteria in Lung Cancer Trials.**

| Types of Exclusion Criteria | % of Lung Cancer Trials (n) |
|---|---|
| Medical History/ Condition | 90.7% (1100) |
| Medical Treatment | 81.5% (924) |
| Clinical Exclusion | 61.6% (699) |
| Gatekeeper Bias | 29.5% (335) |
| Inability to Consent | 5.8% (66) |
| Total | 1,134 Trials |

Note: Percentage totals may not add up to 100% due to some criteria being compounded.

**Table 3. Logistic Regression Predicting Exclusion.**

| Term | Estimate | Odds Ratio | Standard Error | z value | p value |
|---|---|---|---|---|---|
| (Intercept) | -0.58 | 0.56 | 0.09 | -6.73 | p<0.001 |
| Race | | | | | |
| AAPI | 0.57 | 1.76 | 0.06 | 8.95 | p<0.001 |
| American Indian/ Alaska Native | 0.57 | 1.77 | 0.84 | 0.69 | 0.49 |
| Black or African American | 0.46 | 1.59 | 0.11 | 4.08 | p<0.001 |
| Ethnicity | | | | | |
| Hispanic or Latino | 0.05 | 1.05 | 0.15 | 0.33 | 0.74 |
| Sex | 0.00 | 1.00 | 0.00 | 0.22 | 0.83 |
| Male | 0.04 | 1.04 | 0.04 | 0.94 | 0.34 |
| Median Household Income | 0.00 | 1.00 | 0.00 | 0.22 | 0.83 |

pp. The results indicate adequate power for AAPI, moderate power for Black, and insufficient power for Latino participants to detect disparities below 15.8 percentage points.

Frequency analysis was performed to identify common comorbidities among the different racial and ethnic groups in our sample and whether these conditions are commonly included as an exclusion criterion in lung cancer studies. The comorbidities listed excludes lung cancer given that the focus of our study is on lung cancer patients. See Table 4, which also includes disease prevalence data for different demographic groups gathered from existing research.

## Discussion

The need to revisit longstanding clinical trial exclusion criteria for lung cancer studies has been well documented [6]. While strict trial criteria ensure patient safety and reliable data for testing new interventions, when used inflexibly, they can also limit the representativeness of trial populations by disproportionately excluding racial and ethnic minority groups with heavier disease burdens than the general population [10]. To quantify this disparity, our study triangulated data from patient EMR files and lung cancer exclusion criteria from clinicaltrials.gov to assess how these criteria impact eligibility across different racial groups.

Black and AAPI patients were significantly less likely to qualify for enrollment in lung cancer studies based on their existing comorbidities. These findings advance previous research in two important ways. First, while a national survey using self-reported data found that comorbidities reduced clinical trial offers and participation [10], our study, which used EMR data moves beyond previous finding to reveal how specific comorbidities differentially shape trial eligibility across racial and ethnic groups.. Second, by including a racially diverse sample of Black Americans, Asian Americans, and Pacific Islanders (AAPI), we address a critical gap in health disparity research. AAPI participants are typically excluded from such studies and remain the most under-funded and least studied racial and ethnic minority groups in U.S. medical research, leaving substantial gaps in our understanding of health disparities within this population [7,19]. Our findings underscore how AAPI participants may face exclusion from lung cancer clinical trials due to high prevalence of conditions such as diabetes, cardiovascular disease, and cancer—common exclusion criteria for these studies [20].

Notably, we did not find this association for Non-White, Latino patients, despite the fact that Latino patients are significantly underrepresented in lung cancer studies [21], and compared to White patients, disproportionately experience higher disease prevalence of medical conditions that commonly fall under lung cancer exclusion criteria such as type 2 diabetes, hypertension, and serious kidney disease [22–24]. We note that the study had a relatively smaller sample size for Latino American patients compared to AAPI or Black patients. The observed disparities for AAPI (13.6 pp, OR=1.76, p<0.001) and Black (10.7 pp, OR=1.59, p<0.001) patients exceeded the minimum detectable effects for their respective sample sizes, indicating these findings are robust. In contrast, the non-significant finding for Latino patients (1.2 pp,

**Table 4. Top Medical Comorbidities and Prevalence Data by Race and Ethnic Group, and Their Status as Common Exclusion Criteria.**

| Race and Ethnicity | Comorbidities | Listed as a Common Exclusion Criteria in Lung Cancer Studies | Prevalence Data by Race and Comorbidity (Includes Confidence Intervals) |
|---|---|---|---|
| White | | | |
| | Hypertension[1] | Yes | 31.4% (29.7%–33.2%) |
| | Chronic obstructive pulmonary disease (COPD)[2] | Yes | 4.4% (4.1%–4.8%) |
| | Diagnosed diabetes[3] | No | 6.9% (6.6%–7.1%) |
| | Gastroesophageal reflux disease (GERD) | Yes | N/A |
| | Lung disease | Yes | N/A |
| Black | | | |
| | Hypertension[1] | Yes | 45.3% (43.6%–46.9%) |
| | Chronic obstructive pulmonary disease (COPD)[2] | Yes | 3.5% (2.9%–4.2%) |
| | Diagnosed diabetes[3] | Yes | 12.1% (11.3–13.0) |
| | Pulmonary disease | Yes | N/A |
| Asian | | | |
| | Hypertension[1] | Yes | 31.8% (29.9%–33.6%) |
| | Chronic obstructive pulmonary disease (COPD)[2] | No | 1.0% (0.5%–1.6%) |
| | Diagnosed diabetes[3] | Yes | 9.1% (8.2%–10.1%) |
| | Pulmonary disease | Yes | N/A |
| | Liver disease | Yes | N/A |
| Latino | | | |
| | Hypertension[1] | Yes | 31.6% (30.0%–33.2%) |
| | Chronic obstructive pulmonary disease (COPD)[2] | No | 2.0% (1.6% –2.6%) |
| | Diagnosed diabetes[3] | Yes | 11.7% (10.9%–12.6%) |
| | Gastroesophageal reflux disease (GERD) | Yes | N/A |
| | Pulmonary disease | Yes | N/A |

N/A- Prevalence data not available.

[1]See Reference [16].

[2]See Reference [17].

[3]See Reference [18].

OR=1.05, p = 0.74) fell below the minimum detectable effect (15.8 pp), suggesting this null finding may reflect insufficient power rather than definitive evidence of no disparity

Thus, the null findings among Latino patients should be interpreted with caution, as they may be an artifact of limited statistical power given the small sample size within this group. Additionally, the heterogeneity of the Latino patient sample may also mask potential associations that might exist. Latino American population is a heterogenous group composed of different ethnic groups with differential patterns of cardiometabolic diseases, comorbidities, and healthcare access, which are obscured when treating this population as a single analytic category, as most EMR systems typically do. To illustrate, Puerto Rican adults are more likely to have multiple chronic conditions than Central and South Americans [25]. Additionally, the heterogenous genetic backgrounds of different Latino ethnicities, that include Native American, European, and African origin adds to the challenge in defining and examining comorbidities among this group [26,27]. When ethnic information is not collected and tracked, these nuances are less likely to be detected.

Furthermore, our overall results are likely to be a conservative estimate because it does not include clinical exclusion criteria based on laboratory values or vitals, such as lymphocyte count, liver function tests, or neutrophil counts, which are likely to increase a patient's likelihood of not qualifying for a trial, particularly for racial and ethnic minority patients [2,13]. For example,

among patients of African and Middle-Eastern descent, neutrophil counts may be lower than normal, a condition previously called benign ethnic neutropenia, as a result of a genetic variation - Duffy-null phenotype, now called Duffy-null associated neutrophil count (DANC) [28,29]. Though this genetic variation does not indicate a health problem, patients are likely to be excluded because of this variation compared to the general population [30]. According to a recent analyses, standard absolute neutrophil count (ANC) thresholds (e.g., ≥ 1,500/μL) were seen in > 80% of clinical trials testing cytotoxic chemotherapy which could exclude up to 10–20% of individuals with DANC, considering lower reference range of ANC is 1200 [30]. This underscores the need for adjusting clinical trial eligibility such as lowering ANC thresholds and incorporating Duffy null into screening procedures [29,30].

Our study also does not exclude patients based on gatekeeper bias from providers and investigators, resulting in patients being excluded based primarily on the perceiver's surface-level assumptions about patients rather than on standardized clinical observations. Previous research has documented examples of gatekeeping effects resulting from assumptions that providers make about the suitability of racial minority patients [31,32]. For example, some healthcare providers may assume that Asian and Latino patients are not suitable due to cultural differences and language barriers, assumptions that also often affect native English speakers who may be deemed unfit simply based on assumptions of the provider [32]. Thus, the exclusion rate for racial and ethnic minority patients is likely to be higher than what we found, if these factors are taken into account.

### Strengths, limitations, and directions for future research

The strength of our study lies in the triangulation method, which combined data from two independent sources: lung cancer exclusion criteria from clinicaltrials.gov and patient comorbidities data gathered from patient EMR files. Through this effort, we add to the growing body of research illustrating the need for promoting a more adaptable and flexible set of exclusion criteria that accounts for existing health disparities among different racial and ethnic groups to reduce the exclusion of racial and ethnic minority patients who otherwise would benefit from participating in life-saving trials [6,7,33,34].

The study had a few limitations. First, the accuracy and granularity of the classification of the eligibility criteria could be improved. Eligibility criteria in ClinicalTrials.gov's database were semi-unstructured, included complex patterns with varying semantics, and were inconsistent between different trials. Most of the data followed a standard pattern using headings such as "Inclusion Criteria" and "Exclusion Criteria," followed by a numbered or bulleted list of criteria; however, there were entries that veered from that pattern, which may have not fully been accurately classified. For example, many eligibility criteria include exceptions, such as "other than diabetes…", which made classification more difficult. While the research team was able to manually identify "exceptional" languages such as double negatives, there may have been additional variations that our queries excluded. There are also different ways to describe the same disease, with various levels of granularity, meaning there may have been instances where certain diseases were missed. The categorization of eligibility criteria creates an excellent use case to use a large language model that can decipher separate criteria within the layers of complexity. Future research can extend this study by building artificial intelligence (AI) powered large language models that improve the accuracy of categorization and apply similar concepts to other therapeutic areas and other patient samples. It is also important to note that though HIV was a common exclusion criterion in the trial data, we could not obtain EMR records related to HIV for privacy reasons, so this was not included in the analysis. A final limitation relates to the size and nature of racial minority patients in our sample, particularly for our Latino American patient sample. Our study had limited power to detect small-to-medium disparities among Latino participants (n = 72). The minimum detectable effect (15.8 percentage points) exceeded thresholds we considered clinically meaningful (5–10 percentage points). Consequently, we cannot rule out meaningful disparities in this population that our study was underpowered to detect. Future research with larger Latino samples is needed to provide more definitive conclusions. Additionally, our sensitivity analysis was conducted retrospectively using observed data rather than prospectively during study design. While this approach provides valuable information about the interpretability of our findings, ideally power analyses should inform sample size determination before data collection.

## Conclusion

This study provides quantifiable evidence that longstanding lung cancer clinical trial exclusion criteria, particularly those related to comorbidities, may disproportionately limit eligibility for Black and Asian American and Pacific Islander (AAPI) patients. By linking patient-level EMR data with exclusion criteria from ClinicalTrials.gov, our findings demonstrate how routine eligibility requirements systematically disadvantage racial and ethnic groups that bear a higher burden of chronic disease. In doing so, the study advances prior work by identifying specific comorbidities that drive differential exclusion. Our study also explicitly included and measured the impact of clinical trial exclusion on AAPI patients, a population that is often overlooked and remains critically underrepresented in health disparities research [19]. The results underscore how common conditions such as diabetes, cardiovascular disease, and prior cancer diagnoses, despite being common and often manageable, can function as structural barriers to trial participation for different racial groups, highlighting the need for a more flexible application of exclusion criteria interpretability to ensure more equal referral and enrollment of patients.

## Supporting information

**S1 File. S1 eAppendix.** Detailed description of text analysis process.
(DOCX)

## Author contributions

**Conceptualization:** Jennifer Y. Kim.

**Data curation:** Jennifer Y. Kim, Abigail Dirks, Ruby Madison Ford, Lori Pai, Calvin Ludwig, Umit Tapan.

**Formal analysis:** Jennifer Y. Kim, Abigail Dirks.

**Funding acquisition:** Jennifer Y. Kim.

**Investigation:** Jennifer Y. Kim, Abigail Dirks.

**Methodology:** Jennifer Y. Kim, Abigail Dirks, Ruby Madison Ford, Umit Tapan.

**Project administration:** Jennifer Y. Kim.

**Validation:** Abigail Dirks.

**Writing – original draft:** Jennifer Y. Kim, Abigail Dirks, Ruby Madison Ford.

**Writing – review & editing:** Jennifer Y. Kim, Abigail Dirks, Ruby Madison Ford, Lori Pai, Calvin Ludwig, Umit Tapan.

## Acknowledgments

We would like to thank Kyle Zollo-Venecek (CTSI) for his support on this project. CTSI who provided the patient EMR data was supported by the National Center for Advancing Translational Sciences (NCATS) of the National Institutes of Health under Award Number UM1TR004398. The content is solely the responsibility of the authors and does not necessarily represent the official views of the NIH.

We would like to thank Dr. Angelo Williams, DO, an addiction medicine physician who provided input related to the clinical decision-making process that physicians may use when determining patients' ability to consent. Lastly, we would like to thank Lumos Insights for providing methodological support.

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
