## [Decision Letter · Decision Letter 0]

16 Dec 2025

Response to Reviewers
Revised Manuscript with Track Changes
Manuscript

**Journal Requirements:**

1. Please send a completed 'Competing Interests' statement, including any COIs declared by your co-authors. If you have no competing interests to declare, please state "The authors have declared that no competing interests exist". Otherwise please declare all competing interests beginning with the statement "I have read the journal's policy and the authors of this manuscript have the following competing interests:"

1. Please clarify all sources of funding (financial or material support) for your study. List the grants (with grant number) or organizations (with url) that supported your study, including funding received from your institution.

2. State the initials, alongside each funding source, of each author to receive each grant.

3. State what role the funders took in the study. If the funders had no role in your study, please state: “The funders had no role in study design, data collection and analysis, decision to publish, or preparation of the manuscript.”

4. If any authors received a salary from any of your funders, please state which authors and which funders.

**Additional Editor Comments:**
**Reviewers' Comments:**

**Comments to the Author**

1. Does this manuscript meet PLOS Digital Health’s publication criteria?

Reviewer #1: Yes

Reviewer #2: Yes

2. Has the statistical analysis been performed appropriately and rigorously?

Reviewer #1: Yes

Reviewer #2: Yes

3. Have the authors made all data underlying the findings in their manuscript fully available (please refer to the Data Availability Statement at the start of the manuscript PDF file)?

Reviewer #1: Yes

Reviewer #2: No

4. Is the manuscript presented in an intelligible fashion and written in standard English?

Reviewer #1: Yes

Reviewer #2: Yes

Reviewer #1: This manuscript, titled How Exclusion Criteria Can Hinder Eligibility for Lung Cancer Studies Among Different Racial and Ethnic Groups, investigates the impact of clinical trial exclusion criteria on the eligibility of racially and ethnically diverse lung cancer patients in the U.S. The study integrates data from 1,134 U.S.-based lung cancer trials (January 2014–December 2024) sourced from ClinicalTrials.gov and electronic medical records (EMR) of 4,096 lung cancer patients (73.6% White, 12.8% Asian, 3.3% Black, 1.8% Hispanic/Latino) from a Northeast urban academic medical center. However, there are some issues that need to be addressed.

Firstly, the manuscript notes that approximately 16% of exclusion criteria were “compounded” (classified into multiple categories), but it fails to provide details on how these compound criteria were analyzed. For instance, were trials with compound criteria weighted differently? Or were patients excluded if they met any of the multiple criteria categories?

Secondly, Black (3.3%) and Hispanic/Latino (1.8%) patients make up a small fraction of the study sample. Although the authors acknowledge this as a limitation, they do not report a priori power calculations to confirm whether the sample size was sufficient to detect meaningful differences in exclusion rates for these groups. For example, this could include potential associations for Hispanic/Latino patients.

Thirdly, Table 3 lists the top comorbidities by race/ethnicity but does not include prevalence data for these conditions. For example, it does not show whether Type 2 diabetes is significantly more common in AAPI patients than in White patients. Without quantifying the differences in comorbidity prevalence, the causal link between comorbidities and exclusion risk remains indirect. We suggest supplementing Table 3 with comorbidity prevalence rates (and 95% confidence intervals).

Fourthly, the authors note that clinical exclusion criteria (e.g., laboratory values like neutrophil counts) were not included in the analysis, but they do not elaborate on their potential impact on minority groups. For instance, benign ethnic neutropenia, which is common in Black patients, is cited in references 24–25 but is not linked to hypothetical exclusion rates. We recommend adding a paragraph to the “Discussion” section estimating how including these criteria might amplify racial disparities.

Fifthly, there is a critical discrepancy between the title and content of Table 3. The title reads “Types of Exclusion Criteria in Lung Cancer Trials,” but the table actually presents “Top Medical Comorbidities by Race and Ethnic Group.” This error will confuse readers and must be corrected. For example, the title could be revised to “Top Medical Comorbidities by Race and Ethnic Group, and Their Status as Common Exclusion Criteria.” Additionally, ensure that column headers are consistent. For instance, the third column lacks a clear header; it should be “Listed as a Common Exclusion Criterion in Lung Cancer Studies.”

Finally, the authors attribute non-significant results for Hispanic/Latino patients to “sample heterogeneity” but do not explore potential subgroup differences. For example, Mexican American and Puerto Rican patients may have distinct comorbidity profiles. If the EMR data includes subgroup identifiers, adding a post-hoc subgroup analysis of exclusion rates by Hispanic/Latino ethnicity would provide deeper insights into disparities. If not, the “Discussion” section should cite literature on Hispanic/Latino comorbidity heterogeneity to contextualize the null finding.

Reviewer #2: This study addresses an important and timely question: whether clinical trial exclusion criteria disproportionately affect racial and ethnic minority patients with lung cancer. The triangulation of ClinicalTrials.gov data with EMR records is a creative approach, and the topic aligns well with ongoing efforts to improve diversity in clinical research. I appreciate the authors' effort to include AAPI populations, who are often overlooked in health disparities research.

However, I have identified several concerns that require substantial revision before this manuscript is suitable for publication.

Major concerns:

The central claim of the paper is that minority patients are excluded from trials because of their comorbidities. However, this mechanism is assumed rather than demonstrated. The authors do not report comorbidity prevalence by race/ethnicity within their own sample, nor do they show that racial differences in exclusion are mediated by comorbidity burden. Table 3 lists common comorbidities per group but provides no rates, counts, or statistical comparisons. Without this evidence, the association between race and exclusion could reflect unmeasured confounding, site-specific coding practices, or other artifacts. A mediation analysis or at minimum a descriptive table showing differential comorbidity burden by race is essential to support the paper's core argument.

The small sample sizes for Black (3.3%) and Latino (1.8%) patients are acknowledged only briefly at the end of the Discussion. Given that these are central to the study's conclusions, this limitation requires more prominent treatment and should temper the strength of the claims made throughout.

The handling of non-significant findings for Latino patients is problematic. The authors speculate that heterogeneity may be "masking potential associations," but this is post-hoc reasoning without empirical support. If the sample is underpowered or too heterogeneous to detect effects, this should be acknowledged as a limitation rather than reframed as hidden evidence for the hypothesis.

Writing and structure:

The Introduction becomes unclear toward the end, particularly in explaining the study's approach and rationale. This section would benefit from revision for clarity and flow.

The paper lacks a Conclusions section, which is standard for this journal and would help summarize the key findings and implications.

The Discussion at times reads as circular, restating assumptions from the Introduction as if they were demonstrated by the results. The authors should more clearly distinguish between what their data show and what prior literature suggests.

Additional concerns:

The data availability statement indicates that data sharing is subject to funder approval. This is inconsistent with PLOS's open science requirements and should be addressed.

The confidence interval reporting in the Abstract is confusing (e.g., "1.8 (0.57, CI: 1.03-3.03)"). Please clarify and ensure consistency with Table 2.

I recommend a major revision.

**Do you want your identity to be public for this peer review?** For information about this choice, including consent withdrawal, please see our Privacy Policy

Reviewer #1: No

Reviewer #2: No

**Figure resubmission:**

**Reproducibility:** To enhance the reproducibility of your results, we recommend that authors of applicable studies deposit laboratory protocols in protocols.io, where a protocol can be assigned its own identifier (DOI) such that it can be cited independently in the future. Additionally, PLOS ONE offers an option to publish peer-reviewed clinical study protocols. Read more information on sharing protocols at https://plos.org/protocols?utm_medium=editorial-email&utm_source=authorletters&utm_campaign=protocols

---

## [Decision Letter · Decision Letter 1]

7 Feb 2026

How Exclusion Criteria Can Hinder Eligibility for Lung Cancer Studies Among Different Racial and Ethnic Groups

PDIG-D-25-00668R1

Dear Dr. Kim,

We are pleased to inform you that your manuscript 'How Exclusion Criteria Can Hinder Eligibility for Lung Cancer Studies Among Different Racial and Ethnic Groups' has been provisionally accepted for publication in PLOS Digital Health.

Best regards,

Matthew Watson

Guest Editor

PLOS Digital Health

**Additional Editor Comments (if provided):**

Many thanks for your significant efforts in revising your work. Based on reviewer comments, we would now like to accept your manuscript for publication in PLoS Digital Health.

**Reviewer Comments (if any, and for reference):**

Reviewer's Responses to Questions

**Comments to the Author**

Reviewer #2: All comments have been addressed

publication criteria?

Reviewer #2: Yes

3. Has the statistical analysis been performed appropriately and rigorously?

Reviewer #2: Yes

4. Have the authors made all data underlying the findings in their manuscript fully available (please refer to the Data Availability Statement at the start of the manuscript PDF file)?

Reviewer #2: Yes

5. Is the manuscript presented in an intelligible fashion and written in standard English?

Reviewer #2: Yes

Reviewer #2: The authors have made a conscious effort to address all comments and limitations and as consequence the study is greatly improved. I recommend accepting it for publication.

**Do you want your identity to be public for this peer review?** For information about this choice, including consent withdrawal, please see our Privacy Policy

Reviewer #2: No
